# Scoping review of assessment tools for, magnitudes of and factors associated with problem drinking in population-based studies

Kefyalew Dagne ![ORCID] ,[1,2] Bronwyn Myers ![ORCID] ,[3,4] Awoke Mihretu ![ORCID] ,[1] Solomon Teferra[1]

[1]Department of Psychiatry, School of Medicine, College of Health Sciences, Addis Ababa University, Addis Ababa, Ethiopia
[2]Department of Psychiatry, College of Health Sciences and Medicine, Debre Berhan University, Debre Berhan, Ethiopia
[3]Curtin enAble Institute, Faculty of Health Sciences, Curtin University, Perth, Western Australia, Australia
[4]Alcohol, Tobacco and Other Drug Research Institute, South African Medical Research Council, Cape Town, South Africa

**Correspondence to**
Kefyalew Dagne;
kdgc08@yahoo.com

## ABSTRACT

**Background** The term "problem drinking" includes a spectrum of alcohol problems ranging from excessive or heavy drinking to alcohol use disorder. Problem drinking is a leading risk factor for death and disability globally. It has been measured and conceptualised in different ways, which has made it difficult to identify common risk factors for problem alcohol use. This scoping review aims to synthesise what is known about the assessment of problem drinking, its magnitude and associated factors.

**Methods** Four databases (PubMed, Embase, PsycINFO, Global Index Medicus) and Google Scholar were searched from inception to 25 November 2023. Studies were eligible if they focused on people aged 15 and above, were population-based studies reporting problem alcohol use and published in the English language. This review was reported based on guidelines from the 'Preferred Reporting Items for Systematic Reviews and Meta-Analyses extension for Scoping Reviews Checklist'. Critical appraisal was done using the Newcastle-Ottawa Scale.

**Results** From the 14 296 records identified, 10 749 underwent title/abstract screening, of which 352 full-text articles were assessed, and 81 articles were included for data extraction. Included studies assessed alcohol use with self-report quantity/frequency questionnaires, criteria to determine risky single occasion drinking, validated screening tools, or structured clinical and diagnostic interviews. The most widely used screening tool was the Alcohol Use Disorder Identification Test. Studies defined problem drinking in various ways, including excessive/heavy drinking, binge drinking, alcohol use disorder, alcohol abuse and alcohol dependence. Across studies, the prevalence of heavy drinking ranged from <1.0% to 53.0%, binge drinking from 2.7% to 48.2%, alcohol abuse from 4.0% to 19.0%, alcohol dependence from 0.1% to 39.0% and alcohol use disorder from 2.0% to 66.6%. Factors associated with problem drinking varied across studies. These included sociodemographic and economic factors (age, sex, relationship status, education, employment, income level, religion, race, location and alcohol outlet density) and clinical factors (like medical problems, mental disorders, other substance use and quality of life).

**Conclusions** Due to differences in measurement, study designs and assessed risk factors, the prevalence of and factors associated with problem drinking varied widely

across studies and settings. The alcohol field would benefit from harmonised measurements of alcohol use and problem drinking as this would allow for comparisons to be made across countries and for meta-analyses to be conducted.

**Trial registration number** Open Science Framework ID: https://osf.io/2anj3.

---

## STRENGTHS AND LIMITATIONS OF THIS STUDY

⇒ To the authors' knowledge, this is the first scoping review to synthesise the evidence on the prevalence of and factors associated with problem drinking across global settings.
⇒ Strengths include an extensive search of 4 databases, with 81 original articles included for evidence synthesis.
⇒ The review was limited to the community-based studies; studies conducted at institutions like hospitals, primary healthcare centres, addiction centres and colleges or universities were not included.

## INTRODUCTION

The nature of alcohol use, related issues and how they manifest throughout life have long been the subject of scientific research.[1] In 2016, the 'Global Burden of Disease Study' identified alcohol use as a leading risk factor for death and disability, ranking it seventh among the top risk factors for disability-adjusted life years and deaths globally.[2 3] Alcohol use has been identified as a risk factor for more than 200 injuries and diseases, including alcohol use disorder (AUD), liver cirrhosis, malignancies, injuries, tuberculosis, HIV/AIDS,[4 5] non-communicable diseases,[6] mental disorders,[7] violence-related harms and injuries.[8] These problems can arise from acute episodes of alcohol intoxication or chronic, heavy alcohol use.[9] The phrase 'alcohol use disorder' describes the complete range of problematic patterns of alcohol use, ranging from less severe difficulties such

**Table 1** Different definitions and terms for problem alcohol use

| Terms | Definitions |
|---|---|
| Low-risk drinking | Generally defined as a daily intake of no more than 20 g of alcohol with at least two non-drinking days weekly. Low-risk drinking limits are defined differently for cis-gender males and females, that is, not more than three and two drinks a day on average, respectively.[20] |
| Problem drinking (PD) | Problem drinking, commonly referred to as 'alcohol abuse', 'alcohol misuse' or 'AUD', is a pattern of alcohol intake that harms one's health or relationships with others. It is a general term that covers a range of alcohol-related problems, from mild to severe.[11–16] |
| Hazardous drinking | A quantity or pattern of alcohol intake that puts individuals at risk for adverse health events, which carry the possibility of physical or psychological harm.[17 18] |
| Harmful drinking | A quantity and pattern of alcohol intake that causes physical or psychological harm and the presence of physical or psychological complications.[17 19] |
| Heavy episodic/binge drinking (HED/BD) | Defined as the intake of five or more drinks for men and four or more drinks for women per occasion in most studies (roughly 60 g of pure alcohol), which brings blood alcohol concentration (BAC) levels to 0.08 g/dL in about 2 hours.[21] |
| Excessive/heavy drinking (HD) | Heavy drinking is the quantity of alcohol consumed that exceeds a set threshold. It is often defined as the weekly use of more than 14 drinks on average for males and more than seven drinks for females. Some countries define it as the average number of binge episodes per person during 30 days or weekly drinking of more than 21 drinks for males and more than 14 drinks for females.[21–24] |
| Alcohol dependence (AD) | Based on the Diagnostic and Statistical Manual of Mental Disorders—4th edition (DSM-IV), AD is characterised by a problematic pattern of alcohol use that results in clinically significant impairment or distress. It is also a symptom of continuing to use alcohol despite knowing that continued use will cause serious social or interpersonal problems (eg, violent arguments with their spouse while intoxicated or abusing children).[25] |
| Alcohol abuse (AA) | AA is a pattern of alcohol intake that has adverse outcomes and harms a person's physical health, mental health, interpersonal connections and general functioning. AA involves excessive and frequent alcohol consumption despite its harmful effects. It can be less severe than AD because it requires fewer symptoms and can only be diagnosed once the DSM-IV criteria have determined that AD is not present.[25] |
| Alcohol use disorder (AUD) | AUD is a chronic medical disorder defined by an individual's compulsive and problematic pattern of alcohol consumption, diagnosed when an individual's alcohol consumption leads to significant distress or impairment in their daily functioning. It is characterised by a cluster of behavioural and physical symptoms, including withdrawal, tolerance and craving, based on the Diagnostic and Statistical Manual of Mental Disorders—5th edition (DSM-5).[11 26] |

AA, Alcohol abuse; AD, Alcohol dependence; ASSIST, The Alcohol, Smoking, and Substance Involvement Screening Test; AUD, Alcohol use disorder; AUDIT, Alcohol Use Disorder Identification Test; BD, Binge drinking; HD, Heavyy drinking; HED, Heavy episodic drinking; HED/BD, heavy episodic or binge drinking; PD, Problem drinking.

as heavy episodic/binge drinking (HED/BD) and risky drinking to harmful drinking and more serious disorders like alcohol abuse (AA) and alcohol dependence (AD).[10] These different definitions of problem alcohol use and inconsistent ways of measuring these problems have contributed to challenges in understanding the nature and extent of alcohol-related problems across the AUD continuum. In this review, we use the term "problem drinking" to refer to any problem with alcohol use, including AUD. Different definitions and terms for problem alcohol use[11–26] are summarised in table 1.

Alcohol consumption is responsible for a wide range of adverse health outcomes,[3] and alcohol-related harms are well established.[27] Problem drinking, including any form of AUD, is a critical public health issue that has an impact on people and communities all around the world.[28]

Risk factors for the emergence and advancement of problem drinking are not well understood.[2] Despite the severe burden of alcohol use globally, there is fragmented evidence on the contribution of specific risk factors to problem drinking.[2]

Although alcohol consumption occurs on a continuum, our understanding of when to intervene and risk factors to target in interventions is hampered by differences in how problem drinking is conceptualised and measured and the lack of synthesised evidence on factors associated with problem drinking.

A comprehensive global review of evidence on the nature and extent of problem drinking serves several essential purposes. First, it offers crucial epidemiological data, such as burden or prevalence rates, trends and problem drinking patterns over time. With this information, public policy-makers, researchers and healthcare workers may more accurately understand the scope of the problem, pinpoint individuals at high risk and more effectively allocate resources to problem drinking prevention and treatment. Second, the information from the review may be used to create awareness of problem drinking and develop policy initiatives on screening and treatment strategies to reduce its prevalence. Third, studying problem drinking data enables a clearer understanding of factors related to the development or progression of problem drinking. This information is needed to guide prevention initiatives and treatments focusing on specific risk factors, such as the environment, clinical variables and comorbid mental health problems.

Previous reviews recommended a need for further research on the magnitude of problem drinking, focusing on low-income and middle-income countries (LMICs).[2] These reviews targeted specific regions, contexts and populations and focused on a particular type of problem drinking pattern or set of risk factors to the exclusion of others. A review covering a broader range of measures, definitions and associated risk factors will provide a more integrated understanding of the phenomenon, and this will provide an opportunity to identify commonalities and variations of problem drinking across diverse settings and populations.[2]

In summary, this review aims to synthesise the global literature on the nature and extent of problem drinking,

how problem drinking was assessed and factors associated with problem drinking among the general population.

## METHODS

This scoping review was reported based on guidelines from the 'Preferred Reporting Items for Systematic Reviews and Meta-Analyses extension for Scoping Reviews (PRISMA-ScR) Checklist', a tool that is used to guide the scoping review process.[29] A copy of the PRISMA-ScR checklist for scoping reviews is supplemented as an additional file (online supplemental research checklist 1).

### Eligibility criteria

For this review, only articles written in the English language were considered. The PICO framework for prevalence studies (Population, Measurement of presence of disease, Design and Setting) guided the choice of eligibility criteria. Accordingly, for studies to be included, they had to (1) study people aged 15 years or older (Population); (2) report problem drinking or AUD using any screening scales, measures, instruments, clinical diagnostic interviews or laboratory tests to detect alcohol use (Measurement of the presence of disease); (3) have any epidemiological, population-based design (Design); and (4) be located in any country or type of setting, as long as the study had a community-based sample (Setting). Due to the inclusion of all prevalence studies on problem drinking with a global focus and the broad coverage of settings, only population-based studies are included in this scoping review, and studies conducted at primary healthcare centres (PHC), hospital settings, universities or schools are excluded.

### Information sources

The literature search included four databases: PubMed, Embase, PsycINFO and Global Index Medicus and searched from database inception (spanning from 1996, 1974, 1906 and 1948, respectively) to 26 August 2019. Database searching was updated twice: first on 22 July 2022, and second on 25 November 2023. Additional records were identified through other sources, such as Google Scholar.

To ensure methodological rigour, a scoping review protocol for the review was registered with Open Science Framework (OSF), which can be accessed using associated project ID of https://osf.io/2anj3 or registration DOI of https://doi.org/10.17605/OSF.IO/9SYV7.

### Search criteria

The PI (KD) developed the search strategy with close consultations with supervisors (ST and BM). The search strategy consisted of key terms, free texts and controlled vocabulary search terms such as (Medical Subject Heading terms for Medline and Emtree terms for Embase) for the main big terms of "prevalence," "alcohol," and "community/population-based health surveys." Terms within each set were grouped using Boolean 'OR' operators, and terms across sets were combined using 'AND' operators.

Although our scoping review has a global focus, 'Ethiopia' is included as a search term in our search strategy for all databases. Since this scoping review is a formative stage of connected consecutive studies on problem drinking and related alcohol use conditions in Ethiopia and intended to inform further studies, we did not want to miss out on any alcohol-related studies in Ethiopia. Since the Boolean Operator used here is (OR) with the study focus (community/population-based studies), including the term 'Ethiopia' as a search term did not limit the search to studies conducted in Ethiopia or detract from the review's global focus. Terms related to alcohol use and the search strategy for searched databases are included in online supplemental file 1.

### Selection of sources of evidence

After the databases were searched, the titles and abstracts of identified records were imported into EndNote software for deduplication and to facilitate the review process. Two reviewers (KD and AM) independently completed screening article titles and abstracts in the first stage and screening full-text articles in the second stage using a priori inclusion and exclusion criteria to determine eligibility. These two reviewers met to resolve screening and selection differences with discussion and to reach a consensus on whether to include an article. These two independent reviewers assessed the eligibility of 352 full-text articles for the final inclusion of 81 articles in the scoping review. These reviewers achieved a 96.6% level of agreement on which articles to include in the review.

### Data charting process

We developed a data extraction form that included items relating to study characteristics (author, year of publication and citation, study country/location), study design, study setting and population, sample size, study tools or measures and results. Two reviewers (KD and AM) independently extracted data from included studies using this form. These reviewers met to resolve data extraction differences with discussion and to reach a consensus on what to extract from the included articles.

### Collating, summarising and reporting the results

As a scoping review, the aim was to map and aggregate findings to offer and present an overview of the topic and all the material studied. Data were analysed using descriptive statistics, and the results were reported using narrative synthesis and presented in tables.

Although critical appraisal of the quality of included studies is not mandatory in scoping reviews, we decided to assess study quality so that findings from the current scoping review could inform the selection of alcohol screening tools and measures in future studies. We used the 'Newcastle-Ottawa Quality Assessment Scale' for cross-sectional studies.[30–32] We slightly modified the semantics of some items to better align with this review

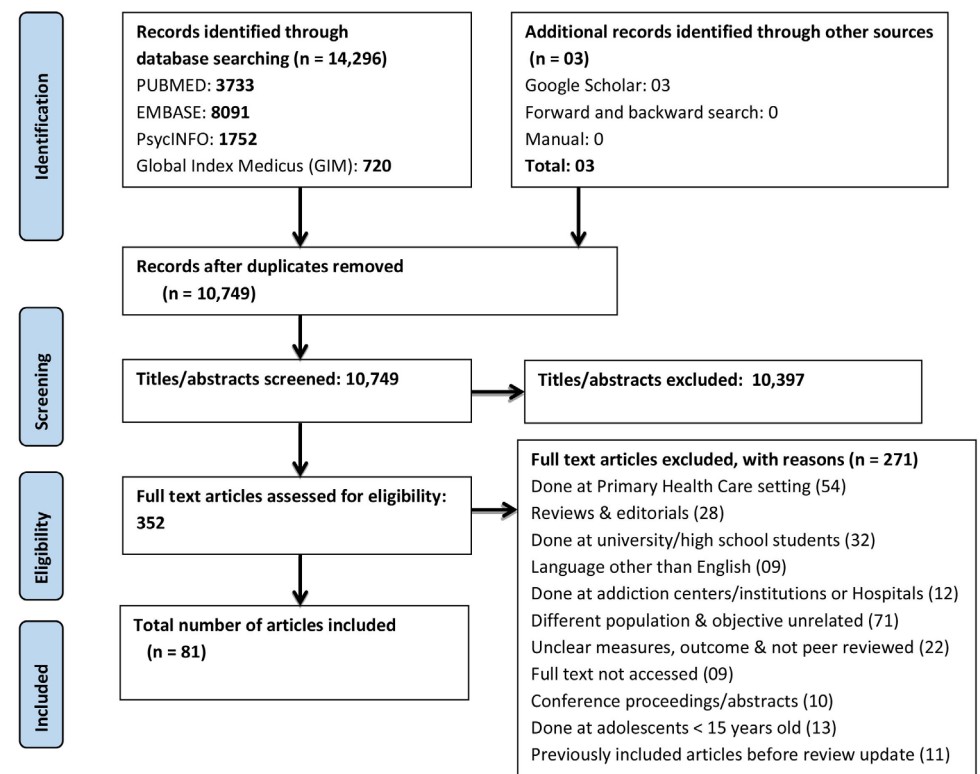

**Figure 1** Preferred Reporting Items for Systematic Reviews and Meta-Analyses flow diagram of included studies in the scoping review, 2023.

(online supplemental file 2). The tool has three domains, each with maximum stars (points/scores): (1) selection (maximum five stars/*****), (2) comparability (maximum two stars/**) and (3) outcome (maximum three stars/***) giving a total score of 10. Studies that scored 9–10 points were considered very good, those that scored 7–8 points were rated as good, those that scored 5–6 points were rated as satisfactory and those that scored 4 points or less were rated as unsatisfactory.[31]

### Patient and public involvement
There was no patient or public involvement in this scoping review.

### RESULTS
The search yielded 14 296 articles from all databases and 3 additional records from Google Scholar. After deduplication, there were 10 749 records, and all these articles underwent title and abstract screening. After titles/abstracts screening, 352 articles were assessed for full-text eligibility, of which 81 articles were included for data extraction. The PRISMA flow diagram summarises this article selection process (figure 1).

### Characteristics of included studies
The publication year for included articles ranged from 1996 to 2023. Only 5 studies were published before 2000, 19 from 2000 to 2010 and 57 from 2011 to 2023. The extracted results of articles from high-income countries (HICs) and LMICs are presented separately in two tables, not for specific purposes but for better visualisation. Of the 81 full-text articles included in this scoping review, 29 were from HICs (online supplemental file 3, table), and the remaining 52 studies were from LMICs (online supplemental file 4, table). Of these 52 studies, 38 were from middle-income countries, 25 were from upper-middle-income countries, 13 were from lower-middle-income countries and 14 were from low-income countries.

Most of the studies employed a cross-sectional study design (73/81), and the rest of the studies were longitudinal/cohort designs (6/81) or mixed quantitative and qualitative designs (2/81). For the majority of included studies (n=30, 37.0%), the study population resided in an urban location, followed by a mixed urban/rural setting (n=27, 33.3% of studies) and rural (n=9, 11.1%). Fifteen (18.5%) studies did not specify the location of the population.

Among the included studies, the total sample size ranged from 99 to 358 355 participants. Only 11 studies had a sample size of less than 500 individuals. Almost 74.1% (n=60) of the studies included had more than 1000 participants in their sample. Nine studies were conducted only among men, two only among women and four studies did not specify gender. Four studies were conducted among young adults (16–25 years old) and seven among older people (adults ≥50 years old). Across studies, participants ranged from 15 to 100 years old, and the mean or median age ranged from 20 to 81.

## Critical appraisal of included studies

When assessing the overall methodological quality of included studies, 17 (21.0%) were rated as very good, 51 (63.0%) as good, 12 (14.8%) as satisfactory and 1 (1.2%) as unsatisfactory (see online supplemental file 5 for quality assessment).

## Measurement of problem drinking

The included studies used a variety of methods to assess problem drinking, including self-report quantity/frequency (QF) questionnaires that included risky single occasion drinking (RSOD) criteria, validated screening tools and structured clinical interviews or assessments (gold standard).

### QF questionnaires and RSOD criteria

Of the 81 included studies, 19 of the 29 conducted in HICs (online supplemental file 3, table) and 21 of the 52 conducted in LMICs (online supplemental file 4, table) have used QF questionnaires. The time interval in which the pattern of alcohol consumption (frequency and quantity) was defined and reported was expressed in days, weeks, months, past 12 months (current use) and ever (lifetime) use. Some studies assessed adherence to country-specific guidelines of recommended limits as part of the QF questionnaires. These guidelines included the French alcohol consumption habits,[33] Australian National Health and Medical Research Council (MRC) 2009 guidelines for mean daily alcohol intake,[34] the Health Council of Netherlands recommended limit for alcohol[35] and the UK National Statistics definition for BD or heavy drinking.[36] Nine studies from HICs (eg, Ireland[37] and Switzerland[38]) and four studies from LMICs applied RSOD criteria. Among HICs, a survey in the US used National Institute on Alcohol Abuse and Alcoholism (NIAAA) guidelines and Substance Abuse and Mental Health Services Administration (SAMHSA) definitions for BD.[39 40]

### Screening and diagnostic interviews for problem drinking

Studies used a variety of screening tools to assess problem drinking. The most commonly used screening tools included the CAGE questionnaire (**C**ut-down on drinking behaviour, **A**nnoyed by criticizing drinking behaviour, **G**uilty feeling about one's drinking, and **E**ye opener first thing in the morning),[41–43] the Alcohol Use Disorders Identification Test (AUDIT),[17] the Michigan Alcohol Screening Test (MAST)[44 45] and the Alcohol, Smoking, and Substance Involvement Screening Test (ASSIST).[46]

Specifically, three studies from HICs[35 47 48] and four from LMICs[49–52] used the CAGE. Five studies from HICs, including New Zealand,[53] the Netherlands,[35] the UK,[54] Norway[55] and Sweden,[56] used either the full or abbreviated versions of the AUDIT. Similarly, 24 studies from LMICs used the AUDIT. The three-item AUDIT-C was used in South Africa, Cambodia, the UK and Sweden,[54 56–58] and a four-item version of the AUDIT—the Fast Alcohol Screening Test (FAST) was used in Ethiopia.[59] Only four studies in LMICs, conducted in Suriname,[60] South Africa[61] and Ethiopia[62 63] used the ASSIST.

The included studies have used five different AUD diagnostic interviews. First, several studies used the Composite International Diagnostic Interview (CIDI).[64–68] Eleven studies from HICs including Hong Kong,[69] Germany,[70 71] Israel,[72] Australia,[73] the Netherlands,[74] Sweden,[75] Ireland,[37] USA,[76] Finland[77] and Switzerland[38] used country-specific versions of CIDI-structured diagnostic tools based on Diagnostic and Statistical Manual of Mental Disorders (DSM)-III, DSM-III-R, DSM-IV, DSM-5 or ICD-10 and ICD-11[19 78] to detect and diagnose AUD, AA or AD. It was also used in three studies from LMICs, including Sri Lanka,[79] Ethiopia[51] and South Africa.[80]

Second, Alcohol Abuse and Alcoholism's Alcohol Use Disorder and Associated Disabilities Interview Schedule-DSM-IV version (AUDADIS-IV)[81] was used in one study in the USA, as HICs.[76]

Third, the Structured Clinical Interview for DSM-IV (SCID-I)[82–84] was used in a Finnish study[77] to detect lifetime DSM-IV substance use disorder.

Fourth, the DSM-IV and DSM-5[25 26] were used by two HIC studies (from Switzerland[38] and Sweden[75]) to diagnose AA, AD or AUD.

Fifth, studies used the Mini International Neuropsychiatric Interview (M.I.N.I.) versions 5, 6 and 7.0.2[85–87] to detect AUD. This is a DSM-IV-based diagnostic tool for detecting AA and dependence during the past 12 months. Only one HIC study (from the USA) used the M.I.N.I.[88] It was employed for the detection of AA or dependence in three studies from LMICs, namely South Africa,[80] Malaysia[89] and Thailand.[90]

## Definitions of problem drinking

Studies defined problem drinking in a variety of ways, including HED/BD, excessive (heavy) drinking or AUD. Definitions of heavy drinking and HED/BD differed according to the recommended drinking limits of countries and how individual studies operationalised the construct. For instance, a study in Finland[47] defined heavy drinking for males as ≥280 g of absolute ethanol or 24 drinks/week and/or a CAGE score ≥3 and for women as ≥190 g of absolute ethanol or 16 drinks/week and/or a CAGE score ≥2. Another study in the USA[39] defined heavy drinking for males as >14 drinks/week and>4 drinks/day and for females as >7 drinks/week and>3 drinks/day. This weekly drinking definition of heavy drinking is also applied in China.[91] A study in France[33] defined heavy drinking as ≥60 g ethanol per day or six glasses per day of any alcoholic drink for males and ≥30 g per day or about three glasses per day for females. Heavy drinking in 2 studies in the Netherlands[35 74] and 1 study in Botswana[24] for women was >14 standard glasses per week, and for men, it was>21 drinks per week. Two studies in Brazil[49 92] operationalised heavy drinking or hazardous drinking as an average of ≥30 g/day, irrespective of gender. Studies from South Africa classified heavy drinking as >7 drinks/week.[93]

HED was sometimes used interchangeably with BD. Studies in Hong Kong[69 94] and the USA[95] defined HED/BD as drinking ≥5 drinks in a row on a single occasion in the past month, irrespective of sex. Most studies described it differently for males and females. The NIAAA guidelines for risky drinking criteria, SAMHSA definition or RSOD criteria were mainly applied to define HED/BD.[93 96–98] In the USA,[76 99] Singapore,[100] Peru,[96] South Africa[57] and Brazil,[97 98 101] HED/BD was defined as ≥5 drinks per occasion for men and ≥4 drinks per occasion for women, a pattern of drinking that brings blood alcohol level to at least 0.08 g/dL and reflects ≥60 g pure alcohol. It was also defined like this by studies conducted in India and Ireland.[37 102] In South Africa, one study[93] used a cut-off of >3 drinks per occasion weekly, and another study[103] used ≥5 drinks on an average drinking day to define HED. Other studies defined HED/BD using different criteria. In Cambodia[58] and Nepal,[104] this was defined as the use of ≥6 drinks in a single sitting at least monthly using NIAAA definitions, and in Ethiopia,[105 106] as an intake of ≥6 drinks in males and ≥4 drinks in females on a single occasion. The definition of BD differed in a study conducted in the UK,[36] with BD defined as >8 standard drinks per session for males and >6 standard drinks per session for females. Some studies examined RSOD, defined as ≥6 drinks per single occasion, and at-risk volume drinking, defined as ≥21 drinks per week, and RSOD at least monthly for men in Switzerland.[38]

Hazardous/harmful alcohol use, also known as harmful/hazardous drinking, probable AUD, risky alcohol use, high-risk drinking, or hazardous, harmful, or dependent alcohol use, was defined as a score of ≥8 on the AUDIT in most studies including studies conducted in New Zealand,[53] Norway,[55] Brazil,[107 108] South Africa,[61 101] India,[109–112] Kenya,[113] Uganda,[114] Nepal,[115] Ethiopia,[63 116–118] Malaysia,[89] Thailand[90 119] and Suriname.[60] This definition is in keeping with the World Health Organization (WHO) recommended cut-offs for problem drinking on the AUDIT.[17] In contrast, one study used an AUDIT score >4 to define hazardous, harmful and high-risk drinking for females in Mozambique.[120]

We noted more variability in the cut-offs used across studies when using short AUDIT forms to define hazardous or harmful drinking. A cut-off score of ≥5 on AUDIT-C (a three-item version of the full AUDIT) was used in South Africa[57] and the UK.[54] Risky drinking was defined as 8–12 for males and 6–12 for females on AUDIT-C in Sweden,[56] while hazardous alcohol use in Ethiopia[59] was defined as a score of ≥3 on the FAST. But a different definition was applied for hazardous drinking in Russia,[121] which was stated as having any of the following in the past year: having drunk surrogate alcohols (non-beverage alcohols and illegally produced alcohols), having been on zapoi (several days of continuous drunkenness during which one withdraws from the society), having frequent hangovers once or more per month and having consumed spirits daily. One study in China[122] used the MAST to define cases of AD, and it was classified using a MAST score of ≥5 with 1–4 (low), 5–6 (light) and 40–53 (severe).

## Prevalence of problem drinking, its pattern and associated factors

### Prevalence and patterns of problem drinking

Six HIC studies assessed heavy drinking (table: online supplemental file 3). Across these studies, the reported prevalence of heavy drinking ranged from 5.0% to 39.9% for males and from <1.0% to 12.9% for females.[33 34 39 47 72] Heavy drinking was reported by 8 out of 47 LMIC studies comprising Brazil,[49 92 97] South Africa,[93 123] Botswana,[24] China[91] and Brazil[52] (table: online supplemental file 4). The prevalence of heavy drinking in these studies ranged from 3.2% to 53.0% in the overall population, 29.2% to 31.0% in males and 3.7% to 17.0% in females.

HED/BD was reported in nine studies conducted in HICs, including Hong Kong,[69] USA,[40 76 95 99] UK,[36] Singapore,[100] Chile[124] and Ireland[37] (table: online supplemental file 3). Across these studies, the prevalence of HED/BD ranged from 14.5% to 24.7% in males, 3.5% to 18.0% in females and 13.7% to 86.0% in the overall sample. HED/BD was also reported by 14 out of 52 studies from LMICs consisting of South Africa,[93 101 103] India,[102] Cambodia,[58] Peru,[96] Brazil,[97 98] Nigeria,[125] Burkina Faso,[126] Nepal[104] and Ethiopia[105 106 116] (table: online supplemental file 4). The overall prevalence of HED/BD ranged from 3.7% to 43.0%. The prevalence of HED/BD ranged from 13.7% to 48.2% in males and 2.7% to 15.0% in females.

The prevalence of AUD, including older diagnostic categories like AA and AD, was reported by 10 out of 29 HIC studies, including Hong Kong,[69] Finland,[77] Germany,[70] Switzerland,[38] Israel,[72] Australia,[73] UK,[54] Sweden,[75] Chicago, USA[88] and Ireland[37] (table: online supplemental file 3). In these studies, the prevalence of any lifetime or current AUD ranged from 4.3% to 36.8% in the overall population, 19.8% to 38.3% in males and 6.3% to 20.6% in females. The prevalence of AA ranged from 4.0% to 4.5%, and AD ranged from 0.4% to 12.3% in the overall sample, 6.1% in males and 6.1% in females.

Likewise, AUD comprising AA, AD, hazardous, harmful or dependent alcohol use was reported by 31 of 52 LMIC studies, including South Africa,[57 61 80 101] Sri Lanka,[79] Ethiopia,[50 51 59 63 116–118] China,[122] Brazil,[49 52 107 108] India,[109–112] Kenya,[113] Uganda,[114] Nepal,[115] Cambodia,[58] Malaysia,[89] Thailand,[90 119] Suriname[60] and Mozambique[120] (online supplemental file 4, table). Either current or lifetime prevalence of any AUD ranged from 4.1% to 41.0% in the overall sample, from 14.5% to 66.6% in males and from 2.0% to 33.4% in females. The prevalence of lifetime or current AA ranged from 6.2% to 9.0% in the overall sample, estimated at 19.0% in males and 6.0% in females. The prevalence of lifetime or current AD ranged from 0.8% to 26.5% in the overall population, from 1.5% to 39.0% in males, and from 0.1% to 19.1% in females.

## Factors associated with problem drinking

Most studies from HICs and LMICs identified factors associated with different types of problem drinking. These factors can be grouped into sociodemographic and socioeconomic; clinical (medical problems or clinical parameters and mental disorders); substance use and risky behaviours; and psychosocial support, functioning, disability and quality-of-life factors (online supplemental file 3, table and online supplemental file 4, table).

Studies from both HICs and LMICs examined a range of sociodemographic factors associated with problem drinking, but the nature and direction of the relationship between these factors and problem drinking were inconsistent. Seven out of 29 studies in HICs found that age was associated with problem drinking. Some studies found that older age was associated with heavy drinking,[35 76] while others found that this association existed for men but not women.[69] In contrast, other studies reported associations between problem drinking and young adulthood,[72 73] with some studies noting that alcohol use declined with age,[56] and age was associated with abstention among women[39] and inversely associated with heavy drinking among men.[33 34] Furthermore, 19 out of 52 studies in LMICs found that age was associated with problem drinking. Some studies reported that older age was associated with alcohol use and different types of problem drinking,[49 51 59 92 101–103 112–115 127 128] while others found that younger age was associated with problem drinking.[58 61 92 96 117 126]

Several studies found associations between male sex and problem drinking. Seven studies from HICs[35 56 70 72 73 76 88] found that male sex was associated with alcohol use and various types of problem drinking. Another 19 studies from LMICs found that male sex was associated with different forms of problem drinking.[24 50 51 57–59 89 92 93 104 105 108 109 113 116–118 126 127]

Some studies from HICs found associations between not being in a relationship and problem drinking, including studies conducted in Australia,[73] Israel[72] and China.[69] Included studies from LMICs also reported associations between not being in a relationship and various types of AUD.[50 60 80 98 102 103 115 123] In contrast, only a handful of studies found that these associations existed for being in a relationship[24 105 120] and age-gap relationships.[24]

In terms of socioeconomic and environmental indicators, only a couple of studies from HICs examined associations between problem drinking and factors like educational attainment,[33 34 74] employment,[69] being immigrants,[72] lower[39] or higher[34] income, location[33 34] or higher neighbourhood alcohol outlet density.[40] Thirteen included studies from LMICs found that education was associated with problem drinking, with some studies finding that a lower educational level was associated with AA and heavy drinking.[49 51 60 101 102 112 121 129] In contrast, others found that this association existed for higher educational levels.[24 61 96 98 128] Thirty-three studies conducted in LMICs examined associations between problem drinking and economic factors, finding

equivocal results. While several studies found associations between lower income[49 50 79 80 92 101 102 127 129] or unemployment[62 121] and problem drinking, others found associations between problem drinking and higher income[57 58 93 101 106 107 109 120 121 127 130] or being employed.[51 58 60 104 106 109 114–116 126 128] Only a few studies from LMICs examined associations between factors like religious affiliation,[50 89 108 128 129] living in urban or rural setting and location[61 101 105 106 112]; ethnicity and race[49 50 57 61 92 93 101 104 115]; household living circumstances[49 103] and problem drinking.

Three studies conducted in HICs[73] and 15 in LMICs[50 59 61 63 79 89 92 97 107 114–118 120] found associations between mental disorders and different forms of problem drinking. Only one HIC study found associations between medical problems like higher body mass index and being non-diabetic than diabetic[39] and problem drinking. In contrast, eight studies from LMICs found associations between medical problems like chronic disease,[63 92] high blood pressure,[91 122] obesity,[93] self-reported physical comorbidities,[112] traffic injury[130] and problem drinking. Only a few studies from LMICs found associations between problem drinking and less psychosocial support,[59 117 118] more impaired functioning, disability, poorer quality of life, cognitive impairment and poor sleep quality.[63 98 111 115 116] In terms of other substance use factors, 7 studies were conducted in HICs,[33–35 69 73 76 77] and 17 studies from LMICs[50 57 61 62 79 92 93 103 105–107 112 115 117 118 126 127] reported associations between cigarette smoking, current khat use, other substance use and various types of problem drinking.

## DISCUSSION

In this scoping review, we identified 81 population-based studies (29 from HICs and 52 from LMICs) that described the prevalence of alcohol consumption and problem drinking and factors associated with problem drinking. Based on the publication year of included articles, there were more than triple the number of published articles in the last decade compared with the previous decade. This increase in publications over time implies that researchers are more interested and involved in alcohol use studies than before.

Despite this growing body of evidence, this review highlights significant heterogeneity of study designs, measures and outcomes that hamper the synthesis of evidence on alcohol prevalence and associated harms across studies. The development of the AUDIT[17] attempted to solve this heterogeneity in the measurement of problem drinking, but the uptake has not been significant.

More specifically, this review identified significant heterogeneity and inconsistency in how various forms of problem drinking were defined and measured,[24 33 35–39 47 49 57 58 69 74 76 91–106] which aligns with previous reviews.[2] Although problem drinking exists on a continuum from mild to more severe, various studies tended to focus on one point in the problem severity

continuum (eg, heavy drinking, HED/BD or AA, AD and AUD) and measures these forms of problem drinking with diverse measurement tools like QF questions, RSOD criteria, screening tools or structured diagnostic interviews.[33–40 46 49–63 69–77 79 80 88–90 93 96–98 101 107–122 128 130] These tools also were variable in the timeframe used to assess problem drinking, with the assessment period ranging from days, weeks, months or years among the studies included in this review.[33–40 62 63 93 96–98 108 128 130]

This variability in how alcohol use and various forms of problem drinking are defined and measured is a significant weakness in the literature, with previous studies noting a lack of attention to the validity of alcohol screening tools and questionnaires.[131] Many challenges in understanding the true prevalence of problem drinking arise from different definitions and inconsistent approaches to measuring it.[2] This was evident in the current review, where we noted considerable differences in the prevalence estimates for problem drinking, partly due to variability in how problem drinking was conceptualised and measured. It is crucial to have a uniform and precise definition of problem drinking that can be applied across studies. This approach will allow for a more accurate estimation of prevalence and more effective identification of people with problem drinking, and it will enhance the robustness of the evidence base on which to advocate for alcohol harm reduction.

Harmonised measures and consensus on the best ways of measuring alcohol use and problem drinking would aid with comparative studies of problem drinking prevalence. Despite the difficulties and challenges associated with building consensus on the best measures for assessing problem drinking and various indicators of problem drinking development, there is an increasing interest in developing agreement on this topic.[132] Notably, even if consensus is reached on which measures of problem drinking to use, these self-report measures would be subject to reporting bias, specifically under-reporting or over-reporting of alcohol consumption. These self-report measures can be supplemented with objective measures of alcohol use (alcohol biomarkers) such as phosphatidylethanol (PEth).[133–138] There is emerging evidence of the benefits of incorporating self-report alcohol use measures with alcohol biomarkers like PEth for valid assessment of problem drinking.[136–149]

Problem drinking is affected by numerous factors at population and individual levels, and identifying these factors is important for informing the design of harm minimisation interventions.[28] The factors associated with problem drinking from our review summarised as sociodemographic and economic characteristics (age, sex, relationship status, education, employment, income level, religion, race, location and alcohol outlet density), clinical factors (medical problems, mental disorders and substance use) and quality of life fit into the biopsychosocial model used in medicine, psychiatry and psychology to understand health and illness.[150 151] This review identified heterogeneity in the kinds of factors that were investigated by included studies as well as the measures used to assess these exposures. This likely contributed to the inconsistent associations found between these factors and the risk of problem drinking.

In addition, it is important to note that this review has weaknesses concerning the examination of factors associated with problem drinking, including the use of less powerful statistical tests (non-parametric tests) or no use of statistical tests,[36 37 47 48 50 53 88 99 110 114 125 152] only a few variables were modelled to control confounding,[71 77 90 96 111 112 124 126] use of non-validated tools that could result in measurement errors,[33 35 36 49 80 94 104 118 128] sampling only (predominantly) males or females that could cause selection bias,[55 63 75 112 120 128] high attrition rates[40 75 129] and small sample sizes.[58 63 89 108 109]

This review highlights the need for additional research on factors associated with problem drinking. Prospective cohort studies that address these methodological limitations and examine the correlates and consequences of problem drinking are needed to guide the design of alcohol harm minimisation interventions. The inconsistency reported in the current scoping review requires a united effort among researchers to refine alcohol use assessment methods to make them clearer and systematise definitions. Hence, future studies could focus on contextual adaptation of WHO-recommended tools like the AUDIT or its shortened versions. Addressing the challenges associated with measuring and defining problem drinking would improve the validity and reliability of future studies, enhance our understanding of the nature and extent of problematic alcohol use, and provide evidence to inform interventions to minimise alcohol-related harms.

### Strengths and limitations

Our scoping review has several strengths. The review protocol was registered at OSF, and we followed PRISMA-ScR guidelines in our scoping review. A comprehensive search strategy was employed to locate global studies. We decided to critically appraise the quality of the included studies, though it is not mandatory in the scoping reviews. This scoping review has several limitations. First, to make our review more feasible, we included only community-based studies, and studies conducted at institutions like hospitals, PHC, addiction centres and colleges/universities were not included, so findings may not be generalisable to these settings. Second, this review was limited to articles published in English. Accordingly, publication bias is possible as studies conducted in other languages and unpublished reports on alcohol use would not have been included.

## CONCLUSIONS

This review highlights heterogeneity in ways in which problem drinking and associated factors have been conceptualised and measured. It also identified methodological weaknesses across the included studies. Together, these findings limit our confidence in the prevalence estimates for problem drinking, our ability to compare findings across studies, and pool data for pooled prevalence estimates. Due to the community-based and cross-sectional nature of the included studies, this review does not provide data on alcohol-related harms. Future alcohol-related research could improve the quality and reliability of findings by strictly following a priori proposed methods and protocols, using validated tools for assessing problem drinking, applying appropriate statistical tests, controlling for possible confounders, minimising selection bias and using a sufficiently large and justifiable sample size.

**Acknowledgements** Our appreciation is dedicated to AMARI (African Mental heAlth Research Initiative) and Addis Ababa University (AAU) for providing training to Kefyalew Dagne in "Systematic Review & Meta-Analysis." The authors would like to acknowledge the Ethiopian Public Health Association (EPHA) Annual Scientific Conference for providing the opportunity to present this research at their 34th (2023) conference.

**Contributors** KD was involved in the project's conceptualisation, writing the protocol, developing a search strategy, searching, screening and extracting included articles, synthesising the results, writing the discussion section of the manuscript and harmonising the entire document. ST approved the conceptualised research project, the protocol and the draft manuscript. BM reviewed the search strategy and provided in-depth reviews of the manuscript. AM was involved in screening and extracting included articles. KD is responsible for the overall content as the guarantor. All authors involved read and approved the final manuscript.

**Funding** Kefylew Dagne was supported through AMARI, funded through the DELTAS Africa Initiative (DEL-15-01). The DELTAS Africa Initiative is an independent funding scheme of the African Academy of Sciences (AAS)'s Alliance for Accelerating Excellence in Science in Africa (AESA) and supported by the New Partnership for Africa's Development Planning and Coordinating Agency (NEPAD Agency) with funding from the Wellcome Trust (DEL-15-01) and the UK government.

**Disclaimer** The views expressed in this publication are those of the author (s) and not necessarily those of AAS, NEPAD Agency, Wellcome Trust, or the UK government.

**Competing interests** None declared.

**Patient and public involvement** Patients and/or the public were not involved in the design, or conduct, or reporting, or dissemination plans of this research.

**Patient consent for publication** Not applicable.

**Provenance and peer review** Not commissioned; externally peer reviewed.

**Data availability statement** All data relevant to the study are included in the article or uploaded as supplementary information.

**ORCID iDs**
Kefyalew Dagne http://orcid.org/0000-0001-7272-6351
Bronwyn Myers http://orcid.org/0000-0003-0235-6716
Awoke Mihretu http://orcid.org/0000-0002-5956-114X

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
