## [Reviewer comments · BMJ Open]

ARTICLE DETAILS

TITLE (PROVISIONAL)	A scoping review of assessment tools for, magnitudes of, and factors associated with problem drinking in population-based studies
AUTHORS	Gizachew, Kefyalew Dagne; Myers, Bronwyn; Awoke, Mihretu; Teferra, Solomon

VERSION 1 – REVIEW

REVIEWER	Madhombiro, Munyaradzi University of Zimbabwe, Department of Psychiatry
REVIEW RETURNED	16-Oct-2023

GENERAL COMMENTS	Abstract The abstract in terms of content covers all aspects of the study. However, the sentences are too long and, in the process, get difficult to follow. For example, line 24-30 on the abstract. The authors need to revise them and make them clearer. Article summary This section is presented well. Introduction Although the introduction gives the rationale of the study. I was particularly curious about the choice of setting especially the exclusion of hospital settings, but this does not come out in the introduction. Will authors clarify this. This is important because individuals with alcohol dependence are more likely to be followed up in specialist centers. In the definition of terms, the authors define alcohol dependence based on the DSM-IV, instead of the current DSM-5? Is there any particular reason, if not, perhaps this can be adjusted. Further, the authors refer to DSM-5 definition when dealing with AUD? In particular reason for this approach? Can the authors clarify. Methods Eligibility criteria This section is written well and clearly Information sources As this manuscript is being reviewed well close to the end of 2023, and yet the searches would have been up to June 2022, is it still current? It would be interesting to update it to current. This will be more work, but the reward is that it will become current and be helpful for all the good reasons for carrying out the study. Search criteria In line 47, I would refer to the supervisors as the senior authors for the purposes of this manuscript, although the relationship would have been different a data collection. Selection of sources of evidence In line 12, it would be important to indicate that the screening was done independently rather than separately. The authors need to
--

	be explicit about how they resolved any differences, was there a tie-breaker? It may need to be stated on how much agreement was there and the measure of agreement. Data charting process This is written clearly. Just a similar point, was there a tie-breaker in case of disagreements? Collating, summarizing and reporting of results. This reads well and authors are applauded for the appraisal done. Results Characteristics of included studies, and critical appraisal of included studies were written well. Definition of problem drinking This section is presented well and clearly. Measures of problem drinking This section is written well and clearly. Prevalence of problem drinking, patterns of, and factors associated The results are presented well. However, with statistical help, wasn't it possible to do some meta-analysis? Given the numbers of the studies, subgroup analysis would have allowed to have some idea of prevalence. The prevalence as given will be difficult to use and ranges are too wide. Discussion The second paragraph appears to be a repetition of the results section. Points raised in the third paragraph are pertinent. However, it has to be noted that WHO in developing the AUDIT, was an attempt at solving this heterogeneity in measurements of PD. The uptake has not been great. Given the findings, what do the authors see as the way forward. Page 19, line 4-19 are repetitions of results section and needs to be revised. It should be contextualizing the findings. Strengths and limitations Conclusion Like noted before, valid alcohol use tools developed for community-based samples are already available. However, these tools are not universally used and attempts at developing more tools based on the current ones such as the AUDIT C is happening. What are the authors suggesting therefore in terms of promoting adoption of tools that are already available. Where would the biomedical screening tools such as phosphatidyl ethanol and their use in conjunction with self-report ones?
--	---

REVIEWER	Thern, Emelie Karolinska Institute
REVIEW RETURNED	20-Nov-2023

GENERAL COMMENTS	Comments to the author General The study aims to synthesize what is known about the assessment of problem drinking, its magnitude, and associated factors. The paper is interesting, well written and highly relevant given that alcohol use can result in tremendous harm and cost, where preventative efforts are needed which is hampered by not measuring alcohol in a harmonized way in research. I do however have some comments and concerns which I have highlighted below: Title and abstract  • In the title the term 'Problem alcohol use' is included but in the aim of the scoping review is to gain more knowledge about the
--

	assessment of 'problem drinking'. Since differences in terminology is a key problem in the current literature, I would suggest that the authors use the same term here and throughout their manuscript. Background  • I am missing a clear rational for the scoping review, what do we know and what does this study add? The way the last sentence in the first paragraph of the background is written it feels like more like the main conclusion of the scoping review as opposed to the gap in the literature. • I also miss a clear justification to why factors associated with alcohol use was included in the scoping review. This has been covered to a greater extent in the literature, and it is not clear to be what this study adds with regards to this. If the hypothesis is that there are differences in the factors associated with alcohol use depending on how problem drink has been measured and defined than this needs to be clarified in the background as well presented in this manner throughout the manuscript. If not, a stronger justification for including this dimension is needed. Method:  • I would suggest including the starting year of when the databases were search, perhaps not all readers know exactly what year all the databases started. I see you have this in the result section but would prefer to have it already in the method section. Results:  • Perhaps something I missed but why are the results presented in terms of high/middle/low-income countries with separate tables – from the aim I am not sure this approach is relevant. And if this is relevant, I suggest adding this to the aim and background to make it clear that this will be how the results will be presented. • A suggestion would be to present the results of how measurement tools of problem drinking first and then the results related to the definitions as this measurement tool often guided how problem drinking has been defined. For me the current order is a bit confusing. Discussion  • As I have previously mentioned I am a bit unsure why associated problems with alcohol were included in the current scoping review. I think the paper could be improved with a stronger justification for this, present the results in relation to how problem drinking was measured and defined (if the hypothesis is that the associated problems could depend on this) and include a more extensive discussion with regards to this. Conclusion  • In relation to my previous comments – I am missing a conclusion in relation to the findings of associated harm in the conclusion. General comment  • Sometimes the abbreviation is used and sometimes not – this is especially in relation to problem drinking but might also be relevant to other abbreviations. To make it easier for the reader please decided on one approach and stick to it throughout the manuscript. Given the number of abbreviations included in the manuscript I would suggest writing out problem drink as opposed to PD just to make it clearer for the reader.
--	--

VERSION 1 – AUTHOR RESPONSE

Reviewer: 1 (Dr. Munyaradzi Madhombiro)

Reviewer's Comment

Abstract

1) The abstract in terms of content covers all aspects of the study. However, the sentences are too long and, in the process, get difficult to follow. For example, line 24-30 on the abstract. The authors need to revise them and make them clearer.

Authors' Response:

Dear Dr. Madhombiro, the comment is very informative, and the lengthy sentence has been revised to be more apparent (Abstract, Page 2).

Article summary

2) This section is presented well.

Authors' Response:

This is well noted

Introduction

3) Although the introduction gives the rationale of the study. I was particularly curious about the choice of setting, especially the exclusion of hospital settings, but this does not come out in the introduction. Will authors clarify this. This is important because individuals with alcohol dependence are more likely to be followed up in specialist centers.

Authors' Response:

Dear Dr. Madhombiro, this is also an important point. It was revised in the last sentence of the introduction as:

Due to the inclusion of all problem drinking prevalence studies globally and the wide coverage of settings, only population-based studies were included in this review, and studies conducted at PHC, hospital settings, universities, or schools were excluded (Page 8).

4) In the definition of terms, the authors define alcohol dependence based on the DSM-IV, instead of the current DSM-5? Is there any particular reason, if not, perhaps this can be adjusted.

Authors' Response:

Thank you Dr. Madhombiro for this comment.

In the DSM-5, the term "alcohol dependence," which has been used in DSM-IV, has been replaced by the broader term "alcohol use disorder."

Since articles included in the review applied all definitions from DSM-III to DSM-5, possible definitions, including the latest (alcohol use disorder based on DSM-5), are provided in the manuscript (Table 1) based on respective DSM definitions to make it more transparent for readers.

5) Further, the authors refer to DSM-5 definition when dealing with AUD? In particular reason for this approach? Can the authors clarify.

Authors' Response:

Thank you for the question again, Dr.

The clarification for a comment on number (4) above applies similarly to this comment.

Methods

Eligibility criteria

6) This section is written well and clearly

Authors' Response:

Well noted and thank you.

Information sources

7) As this manuscript is being reviewed well close to the end of 2023, and yet the searches would have been up to June 2022, is it still current? It would be interesting to update it to current. This will be more work, but the reward is that it will become current and be helpful for all the good reasons for carrying out the study.

Authors' Response:

Dear Dr. Madhombiro, this is an important suggestion. Based on your recommendation, I have updated the literature search until November 25, 2023, screened results, extracted data, and incorporated it with an existing manuscript. Details of all added results, modified PRISMA flow diagram, and search strategy are all marked and updated (Page 8 and 9).

Search criteria

8) In line 47, I would refer to the supervisors as the senior authors for the purposes of this manuscript, although the relationship would have been different a data collection.

Authors' Response:

Well noted, and we accept your comment, Dr. Madhombiro

Selection of sources of evidence

9) In line 12, it would be important to indicate that the screening was done independently rather than separately. The authors need to be explicit about how they resolved any differences, was there a tie-breaker? It may need to be stated on how much agreement was there and the measure of agreement.

Authors' Response:

Thank you, Dr. Madhombiro, for these crucial comments.

"Separately" is replaced by "independently." A statement is modified to clarify how tiebreaker was handled: "These two reviewers met to resolve screening and selection differences with discussion and to reach a consensus on whether to include an article." The discussion between two reviewers to resolve disagreements was a "tiebreaker" here.

The measure of agreement is incorporated in the revised manuscript in the "Selection of sources of evidence" section (Page 9).

Data charting process

10) This is written clearly. Just a similar point, was there a tie-breaker in case of disagreements?

Authors' Response:

The discussion between the two reviewers to resolve disagreements was also considered a "tiebreaker" here (Page 10).

Collating, summarizing and reporting of results.

11) This reads well and authors are applauded for the appraisal done.

Authors' Response:

Thank you for the compliment.

Results

12) Characteristics of included studies, and critical appraisal of included studies were written well.

Authors' Response:

Well noted and thank you.

Definition of problem drinking

13) This section is presented well and clearly.

Authors' Response:

Well noted and thank you.

Measures of problem drinking

14) This section is written well and clearly.

Authors' Response:

Thank you for the compliment.

Prevalence of problem drinking, patterns of, and factors associated

15) The results are presented well. However, with statistical help, wasn't it possible to do some meta-analysis? Given the numbers of the studies, subgroup analysis would have allowed to have some idea of prevalence. The prevalence as given will be difficult to use and ranges are too wide.

Authors' Response:

Dear Dr. Madhombiro, you have raised another important point.

Apart from many numbers of studies, we would not expect a meta-analysis to be part of a scoping review. Since this scoping review aims to map the existing literature on a broad topic (problem drinking), measures and definitions are highly heterogeneous. Pooling prevalence data would be difficult even with sub-group analysis. Meta-analysis would be appropriate if our aim was a systematic review focusing on a very narrow (specific) topic.

Discussion

16) The second paragraph appears to be a repetition of the results section.

Authors' Response:

This is also an interesting point. This paragraph is modified as follows: "Based on the publication year of included articles, there were more than triple the number of published articles in the last decade compared to the previous decade." This implies that researchers are interested and involved in alcohol use studies to date (Page 19).

17) Points raised in the third paragraph are pertinent. However, it has to be noted that WHO in developing the AUDIT, was an attempt at solving this heterogeneity in measurements of PD. The uptake has not been great.

Authors' Response:

Thank you, Dr. Madhombiro, for your insightful comment. I have incorporated your feedback as: "The development of AUDIT with the WHO collaborative project (17) attempted to solve this heterogeneity in measurements of problem drinking, but the uptake has not been significant" (Page 19).

18) Given the findings, what do the authors see as the way forward.

Authors' Response:

Based on your informative feedback, the way forward and implications of the review are added at the end of the discussion (Page 21 & 22).

19) Page 19, line 4-19 are repetitions of results section and needs to be revised. It should be contextualizing the findings.

Authors' Response:

Thank you for the observation, which helps the manuscript improve. Repetitions are corrected (Page 20).

Strengths and limitations

Conclusion

20) Like noted before, valid alcohol use tools developed for community-based samples are already available. However, these tools are not universally used and attempts at developing more tools based on the current ones such as the AUDIT C is happening. What are the authors suggesting therefore in terms of promoting adoption of tools that are already available.

Authors' Response:

This is another excellent recommendation, Dr. Madhombiro. Apart from the way forward, the need for the adoption/adaptation of WHO-recommended tools like AUDIT or its abbreviated versions are provided (modified on the manuscript, at last sentences of the discussion, Page 22).

21) Where would the biomedical screening tools such as phosphatidyl ethanol and their use in conjunction with self-report ones?

Authors' Response:

This is an important point to be informative on alcohol biological markers and modified with supporting references as:

"as there are emerging studies focused on incorporating self-report alcohol use measures with alcohol biomarkers like PEth for valid assessment of problem drinking (137-150)." (Page 21).

Reviewer: 2 (Dr. Emelie Thern)

Comments to the Author:

General

1) The study aims to synthesize what is known about the assessment of problem drinking, its magnitude, and associated factors. The paper is interesting, well written and highly relevant given that alcohol use can result in tremendous harm and cost, where preventative efforts are needed which is hampered by not measuring alcohol in a harmonized way in research. I do however have some comments and concerns which I have highlighted below:

Authors' Response:

Dear Dr. Thern, thank you for your encouraging statement about our study; we have tried our best to provide a point-by-point response for your comments below.

Title and abstract

2) In the title the term 'Problem alcohol use' is included but in the aim of the scoping review is to gain more knowledge about the assessment of 'problem drinking'. Since differences in terminology is a key problem in the current literature, I would suggest that the authors use the same term here and throughout their manuscript.

Authors' Response:

Your feedback is an essential input for our manuscript, Dr. Thern. The term 'Problem alcohol use' is replaced by 'problem drinking', and this latter term is used consistently in the whole document.

Background

3) I am missing a clear rationale for the scoping review, what do we know and what does this study add?

Authors' Response:

It's really an important comment, Dr. Thern. Clear rationale/justification, what is known, the gaps, and what the study adds are incorporated in the introduction section (Pages 6, 7, & 8).

4) The way the last sentence in the first paragraph of the background is written it feels like more like the main conclusion of the scoping review as opposed to the gap in the literature.

Authors' Response:

Thank you, Dr. Thern, for your critical look. The sentence is modified accordingly in a way that conveys a gap in the literature, and it is moved to other similar concepts (below Table 1, Page 6/7).

5) I also miss a clear justification to why factors associated with alcohol use was included in the scoping review. This has been covered to a greater extent in the literature, and it is not clear to be what this study adds with regards to this. If the hypothesis is that there are differences in the factors associated with alcohol use depending on how problem drink has been measured and defined than this needs to be clarified in the background as well presented in this manner throughout the manuscript. If not, a stronger justification for including this dimension is needed.

Authors' Response:

Your feedback is important, and we addressed it as recommended as it improves the manuscript. Clear and robust justification is provided as to why factors associated with problem drinking are included in the scoping review results in the introduction/background section (Page 7).

Method:

6) I would suggest including the starting year of when the databases were search, perhaps not all readers know exactly what year all the databases started. I see you have this in the result section but would prefer to have it already in the method section.

Authors' Response:

Thank you for the recommendation, Dr. Thern.

Starting year (inception) of databases are added and modified on the main manuscript under the sub-section 'Information sources' (Page 8).

Results:

7) Perhaps something I missed but why are the results presented in terms of high/middle/low-income countries with separate tables – from the aim I am not sure this approach is relevant. And if this is relevant, I suggest adding this to the aim and background to make it clear that this will be how the results will be presented.

Authors' Response:

We appreciate your comment on the clarity issue with a separate presentation of tables.

This was due to inclusion of many studies (81 articles) and fitting all extracted data on one table is very difficult (table for HICs: 7 pages, and table for LMICs: 15 pages). In addition, readers can easily go and grasp what they need for the specific table as each table is referenced/cited separately on the main document of the paper. Results of each HICs and LMICs were written separately in the first draft, but with the senior supervisors' suggestions, it was merged side to side for easier interpretation or comparison.

The presentation approach is clarified in the introduction section next to aim, as suggested (Page 8).

8) A suggestion would be to present the results of how measurement tools of problem drinking first and then the results related to the definitions as this measurement tool often guided how problem drinking has been defined. For me the current order is a bit confusing.

Authors' Response:

Your suggestion is informative. To avoid confusion, measures and definitions are swapped over (Pages 11-15).

Discussion

9) As I have previously mentioned I am a bit unsure why associated problems with alcohol were included in the current scoping review. I think the paper could be improved with a stronger justification for this, present the results in relation to how problem drinking was measured and defined (if the hypothesis is that the associated problems could depend on this) and include a more extensive discussion with regards to this.

Authors' Response:

Dear Dr. Thern, your recommendation is highly appreciated.

Based on your previous comment, a clear and robust justification is provided as to why factors associated with problem drinking are included in results in the introduction/background section (Pages 7).

Unfortunately, we didn't put a hypothesis that the associated problems could depend on measures and definitions of problem drinking since we aimed to map and synthesize the existing literature on problem drinking. A discussion of the synthesized factors with pre-existing models and relationship with measures/definitions is provided (Page 21).

Conclusion

10) In relation to my previous comments – I am missing a conclusion in relation to the findings of associated harm in the conclusion.

Authors' Response:

Thank you again for suggesting addressing the considerable impact of alcohol use and alcohol-related harms. The conclusion is modified as:

Due to the cross-sectional and community-based nature of the included studies, findings of alcohol-related harm are missing in our review, which is our target area in our next longitudinal studies (Page 23).

General comment

11) Sometimes the abbreviation is used and sometimes not – this is especially in relation to problem drinking but might also be relevant to other abbreviations. To make it easier for the reader please decided on one approach and stick to it throughout the manuscript. Given the number of abbreviations included in the manuscript I would suggest writing out problem drink as opposed to PD just to make it clearer for the reader.

Authors' Response:

This comment is another great piece of feedback that improves the clarity of our paper. Instead of PD, 'problem drinking' is substituted and used throughout the document, tables, figures, and supplementary materials.

For alcohol use disorder (AUD), the abbreviation (AUD) is used consistently after it's first use written in full text. Heavy drinking abbreviated as HD is kept being used as "heavy drinking" to avoid confusion with heavy episodic drinking (HED) since these two terms have different definitions. Alcohol abuse (AA) and alcohol dependence (AD) are also used in a full-text form consistently except for first

use with abbreviations to avoid confusion. Other abbreviations are also checked to be used consistency.

(All definitions provided in a Table 1)

VERSION 2 – REVIEW

REVIEWER	Madhombiro, Munyaradzi University of Zimbabwe, Department of Psychiatry
REVIEW RETURNED	29-Dec-2023

GENERAL COMMENTS	Thank you for addressing my concerns. However, please attend to minor grammar. Further, I would recommend the use of DSM 5, which is already 10 years in your definitions.
--

REVIEWER	Thern, Emelie Karolinska Institute
REVIEW RETURNED	02-Jan-2024

GENERAL COMMENTS	Great work in meeting my comments, I have nothing further to add.
---

VERSION 2 – AUTHOR RESPONSE

Reviewer 1: (Dr. Munyaradzi Madhombiro)

Reviewer's Comment

1) Thank you for addressing my concerns. However, please attend to minor grammar. Authors' Response:

Dear Dr. Madhombiro, thank you for acknowledging that your previous concerns have been addressed. Based on your feedback on the grammar check, a senior native English-speaking supervisor had a good read and made quite a number of edits to improve the English language and grammar. All other authors were involved in the final read and review of the entire manuscript and related documents. All revisions are found in the main document (marked copy) of the current R2 submission.

2) Further, I would recommend the use of DSM 5, which is already 10 years in your definitions.

Authors' Response:

Dear Dr. Madhombiro, this is an excellent recommendation for us to use the latest definitions of DSM-5. The reason definitions of other DSM versions (DSM-IV) are employed in the manuscript is that the search year was not restricted during our search. All databases from inception until now have been searched, and old papers using old definitions like alcohol dependence (AD) and alcohol abuse (AA) were reported in some papers. Accordingly, to make readers aware of these terms used by some

previous articles in our syntheses, definitions according to old versions of DSM-IV were intentionally included. Your comment is highly appreciated, and we are applying the new DSM-5 definitions in our ongoing original studies and will apply DSM-5 in our future studies too.

Reviewer: 2 (Dr. Emelie Thern)

Comments to the Author:

1) Great work in meeting my comments, I have nothing further to add.